# Nutraceutical Prospects of Pumpkin Seeds: A Study on the Lipid Fraction Composition and Oxidative Stability Across Eleven Varieties

**DOI:** 10.3390/foods14030354

**Published:** 2025-01-22

**Authors:** Magdalena Grajzer, Weronika Kozłowska, Iwan Zalewski, Adam Matkowski, Justyna Wiland-Szymańska, Monika Rękoś, Anna Prescha

**Affiliations:** 1Department of Dietetics and Bromatology, Wroclaw Medical University, Borowska 211, 50-556 Wroclaw, Poland; iwan.zalewski@umw.edu.pl (I.Z.); anna.prescha@umw.edu.pl (A.P.); 2Division of Pharmaceutical Biotechnology, Department of Pharmaceutical Biology and Biotechnology, Wroclaw Medical University, Borowska 211, 50-556 Wroclaw, Poland; weronika.kozlowska@umw.edu.pl; 3Botanical Garden of Medicinal Plants, Division of Pharmaceutical Biology and Botany, Department of Pharmaceutical Biology and Biotechnology, Jana Kochanowskiego 14, 50-367 Wroclaw, Poland; bbsekret@umw.edu.pl; 4Department of Systematic and Environmental Botany, Faculty of Biology, Adam Mickiewicz University, Uniwersytetu Poznańskiego 6, 61-614 Poznan, Poland; wiland@amu.edu.pl; 5Botanical Garden, Adam Mickiewicz University, Dąbrowskiego 165, 60-594 Poznan, Poland; monika.rekos@amu.edu.pl

**Keywords:** lipid fraction, *Cucurbita maxima*, *Cucurbita moschata*, *Cucurbita pepo*, oxidative stability

## Abstract

The oxidative stability of nutritive and bioactive lipids is essential for their functionality. This study evaluated the potential of lipid fractions from pumpkin seeds obtained from eleven high-performing cultivars of *Cucurbita maxima* Duchesne, *C. pepo* L., and *C. moschata* Duchesne cultivated in Poland, aiming to evaluate their stability for nutraceutical applications. This study investigated the intrinsic relationship between chemical composition and oxidative stability to identify cultivars with promising functional potential and commercial value. The fatty acid, sterol, and lipid antioxidant profiles were characterized using gas chromatography (GC), GC–mass spectrometry (GC-MS), and ultra-high-performance liquid chromatography (UPLC), respectively. Antiradical activity was assessed via the DPPH assay, and oxidative stability was evaluated using differential scanning calorimetry (DSC). The oils exhibited high levels of polyunsaturated fatty acids (PUFAs) (59.5–68.6%), with n-6/n-3 fatty acid ratios ranging from 66.5 to 211.6. The lipid extracts contained up to 97.1% Δ^7^-sterols, while key antioxidants included squalene (616.6–3092.0 mg/kg) and γ-tocopherol (54.1–423.6 mg/kg). Notably, the *C. pepo* cultivar ‘Moonshine’ was the least abundant in these bioactive compounds. The carotenoid content ranged from 5.7 to 19.4 mg/kg across the extracts. Among the studied cultivars, ‘Show Winner’ and ‘Pink Jumbo Banana’ (*C. maxima*) stood out as promising candidates for nutraceutical applications due to their elevated levels of tocopherols, carotenoids, and squalene. A moderate n-6/n-3 fatty acid ratio (100–170), coupled with balanced levels of γ-tocopherol and squalene, was found to significantly enhance the oxidative stability of pumpkin seed lipids. These lipid fractions also show potential as stabilizing additives for oils rich in α-linolenic acid but deficient in natural antioxidants.

## 1. Introduction

Pumpkin seeds contain an impressive lipid fraction, averaging about 44% of fat content, though this varies from 31 to 48.6% depending on the cultivar [1,2,3]. Sourced primarily from *Cucurbita maxima* Duchesne, *C. pepo* L., and *C. moschata* Duchesne, the lipid fraction offers valuable nutraceutical properties. This lipid fraction is distinguished by its high antioxidant activity, offering substantial health benefits, such as helping to prevent hypertension and certain cancers. Additionally, the consumption of pumpkin seed oil has also been associated with the inhibition of arthritis development, glycemic regulation in individuals with diabetes, and anti-inflammatory effects. It also exhibits diuretic properties and has shown potential for alleviating symptoms of prostate hyperplasia [2,4,5,6].

Approximately 70% of unsaturated fatty acids in pumpkin seed oil are unsaturated [7], predominantly oleic acid (OA) (17–44%) and linoleic acid (LA) (35–63%) [1,2,4,6]. The composition of unsaturated fatty acids is largely responsible for the oil’s hypocholesterolemic effect and plays a key role in enhancing its oxidative stability, which is crucial for its potential use in nutraceutical formulations [8]. Pumpkin seed oil may contain a substantial range of phytosterols (128–2087 mg/100 g), with a unique profile rich in Δ7-sterols, unlike most vegetable oils, which are primarily composed of Δ^5^-sterols [2,6,9]. Among these Δ^7^-sterols are specific compounds, such as spinasterol, Δ^7^-stigmasterol, Δ^7,25^-stigmastadienol, and Δ^7,22,25^-stigmastatrienol, known for their potential to support prostate health by preventing excessive cell proliferation, while Δ^7^-avenasterol, also present in this group of compounds, is recognized for its free radical stabilization capabilities, particularly through the formation of stable tertiary free radicals at the C24 position [10].

Additionally, the presence of squalene (approximately 650 mg in 100 g of oil) contributes to its anticancer properties [10,11]. The oil is also rich in tocopherols (18–88 mg/100 g) and carotenoids (approx. 3.9 mg/100 g), which further support its role in oxidative stability and health promotion, making it a valuable addition to nutraceutical formulations [12,13].

Given that the composition of pumpkin seed oil is influenced by the species, cultivar, and climatic and agrotechnical conditions, as well as the extraction method, selecting varieties that are rich in bioactive compounds, resistant to oxidation, and suitable for nutraceutical applications in a specific region of the world is a challenging task. This study hypothesizes that among the eleven cultivars of pumpkin (*C. maxima* Duchesne, *C. pepo* L., and *C. moschata* Duchesne), those producing the highest yields under Polish climatic conditions will exhibit variations in lipid composition, enabling the identification of the best-performing varieties. Specifically, this research aims to select cultivars that are most effective in stabilizing oxidation and prolonging shelf life. Such selected cultivars could also serve as raw materials for the production of additives to other edible and cosmetic oils, as well as lipid fractions derived from natural sources. These additives could provide targeted bioactive effects while compensating for deficiencies in certain components, including those critical for ensuring stability. Additionally, blending these lipid fractions with oils rich in alpha-linolenic acid (ALA), such as flaxseed oil or rose hip seed oil, could not only enhance the oxidative stability of these blends but also improve their nutritional value. This enhancement could be achieved by introducing linoleic and oleic acid and bioactive compounds unique to pumpkin seed oil, such as Δ^7^ -phytosterols and squalene, thereby mitigating the pro-inflammatory potential of ALA-rich oils with a low content of more stable fatty acids.

This study aims to investigate the relationship between the composition of lipid extracts from pumpkin seeds and their oxidative stability. By identifying the optimal composition of these extracts, this research seeks to enhance their resistance to oxidation, paving the way for improved nutraceutical applications and sustainable use of plant-based lipid sources.

## 2. Materials and Methods

### 2.1. Materials

Eleven pumpkin cultivars of three species, namely *C. maxima* Duchesne, *C. moschata* Duchesne, and *C. pepo* L., were used. The seedlings were obtained from certified seeds and represented the following varieties: *C. maxima* ‘Atlantic Giant’, *C. maxima* ‘Da Marmellata’, *C. maxima* ‘Golden Hubbard’, *C. maxima* ‘Pink Jumbo Banana’, *C. maxima* ‘Rouge Vif d’Etampes’, and *C. maxima* ‘Show Winner’; *C. pepo* ‘Moonshine’, *C. pepo* ‘Mustang F1’, *C. pepo* ‘Jack Sprat F1’, and *C. pepo* ‘Oblonga’; and *C. moschata* ‘Butternut Rugosa’. These cultivars were selected based on their ability to produce high seed yields under Polish climatic conditions, a factor that is not achievable for all pumpkin cultivars. The seeds were planted in outdoor plots in the Botanical Garden of the Adam Mickiewicz University in Poznan (Poland, 52°25′11.70″ N 16°52′55.07″ E) in common garden conditions. Plants were brought to maturity, and then the fruits were manually collected. The seeds were manually extracted from ripe pumpkin fruits, manually cleaned of pumpkin pulp, and air-dried under a roof in a glasshouse without heating, at ambient temperatures not exceeding 24 °C.

All chemicals used in this study were of analytical grade and obtained from Merck (Merck KGaA, Darmstadt, Germany), unless specified otherwise.

### 2.2. Methods

#### 2.2.1. Extraction of Lipids from Pumpkin Seeds

Seeds were dehulled and then lipids were extracted with n-hexane according to Rezig et al. [2]. A portion of freeze-dried, dehulled, crushed seeds (50 g) was transferred to a dark glass bottle, poured over with 250 mL of n-hexane, and shaken under nitrogen for 4 h at room temperature. Freeze drying was conducted by freezing whole pumpkin seeds at −80 °C for 24 h, followed by sublimation of ice under vacuum at 0.01 mbar and −50 °C until a constant weight was achieved. This ensured the complete removal of water, preserving the stability and composition of bioactive compounds, such as tocopherols, carotenoids, and phytosterols.

The Soxhlet method was chosen as it is widely regarded as a benchmark for lipid extraction due to its reliability and efficiency in isolating non-polar bioactive compounds from complex plant matrices [14]. The extraction procedure was repeated with another portion of n-hexane, and then the extracts were combined. The obtained extract was filtered through a quality paper filter into round-bottomed flasks previously weighed to the nearest 0.001 g and then concentrated to a constant weight on a rotary evaporator in a nitrogen atmosphere at 30 °C. The flask with the residue from the evaporation of the solvent was weighed on an analytical balance, and the lipid fraction was transferred to an amber glass bottle filled with gaseous nitrogen. Lipid extracts prepared in this way were used for further analyses.

#### 2.2.2. Fatty Acid Composition Analysis

The composition of fatty acids in the tested lipid fractions was determined by gas chromatography according to the method of Prescha et al. [15], using a capillary column (Supelco SPTM-2560 fused silica capillary column, 100 m × 0.25 mm × 0.2 μm, Bellefonte, PA, USA) to separate volatile fatty acid methyl esters (6890 N gas chromatograph, Agilent Technologies, Santa Clara, CA, USA). The identification and relative quantification were performed using an external standard mixture of fatty acid methyl esters (Supelco, Bellafonte, PA, USA, 37 Component FAME Mix, CRM47885).

#### 2.2.3. Phytosterol and Squalene Identification and Determination

Lipid fractions for the determination of phytosterols and squalene were prepared using the method of Shukla et al. [16]. The composition of phytosterols and squalene was determined by gas chromatography with a capillary column, following prior derivatization of sterols to silyl derivatives. The analysis was performed on a Clarus 680 gas chromatograph coupled with a Clarus SQ 8 T mass spectrometer (PerkinElmer, Waltham, MA, USA). Silyl sterol derivatives and squalene were separated on a non-polar HP-1 MS 100% Dimethyl Siloxane capillary column (30 m × 250 μm × 0.25 μm, Agilent, Santa Clara, CA, USA). The injection volume of the sample on the column was 2 μL. Hydrogen was used as the carrier gas at 21.9 psi at 40 mL/min, air was used as the make-up gas at 450 mL/min, and nitrogen was used as the masking gas at 45.0 mL/min. The detector temperature was 310 °C. The chromatographic separation was carried out in a temperature program with an initial temperature of 250 °C for 5 min and then an increase of 5 °C/min to the final temperature of 290 °C, which was maintained for 13.5 min [17]. Additionally, pumpkin seed oil samples were analyzed using an Elite-17 MS 50% Phenyl/50% Methyl polysiloxane (30 m × 0.25 mm × 0.25 μm, PerkinElmer, Waltham, MA, USA) capillary column to separate the β-sitosterol and spinasterol peaks that coeluted on the HP-1 column. The temperature program used for the chromatographic separation was as follows: initial temperature of 70 °C for 1.5 min, rise of 40 °C/min to 245 °C held for 1.5 min, and then rise of 2 °C/min to 280 °C held for 10 min. The helium flow through the column was 1.5 mL/min [18]. The identification of individual compounds from the sterol group was carried out through analyses of mixtures of available reference substances and using the NIST Search 2.0 library for compounds without an available reference substance (campesterol, Δ^7,22,25^-stigmastatrienol, Δ^7,25^-stigmastadienol, Δ^7^-stigmastenol). The internal standard method with 5α-cholestane was used for quantitative analyses.

#### 2.2.4. Tocopherol and Carotenoid Identification and Determination

The analytical procedure for tocopherols and carotenoids involved the extraction of the unsaponifiable fraction of lipid samples, followed by identification and quantification using UPLC. To mitigate matrix effects, lipid samples were saponified and then extracted prior to chromatographic analysis. Saponification was conducted at room temperature in the presence of ascorbic acid, which was added to the sample before the procedure to prevent the oxidation of sensitive tocopherols and carotenoids. This step effectively removed triacylglycerides and chlorophylls from the oil samples.

The saponification and extraction procedure followed the method established by Fromm et al. [19]. Specifically, 0.5 g of lipid sample was placed into a 100 mL glass bottle with 40 mL of 12% potassium hydroxide in 80% methanol and 0.25 g of ascorbic acid. The addition of water to methanol inhibited the transesterification and methylation of fatty acids. After adding the reactants, the bottles were flushed with nitrogen gas, sealed, and gently shaken for 24 h at room temperature.

Following saponification, 40 mL of deionized water was added, and the mixture was transferred to a separatory funnel. The unsaponifiable fraction was extracted with a 40 mL hexane–ethyl acetate mixture (85:15, *v*/*v*). The hexane layer was collected, and the extraction was repeated three additional times to maximize recovery. The combined organic fractions were washed with deionized water to neutral pH, dried over anhydrous sodium sulfate, and filtered through qualitative filter paper into round-bottom flasks. Solvents were evaporated under vacuum at 30 °C, and the dry residue was dissolved in 1 mL of isopropanol. Samples were stored under nitrogen in amber glass vials at −80 °C until analysis.

For quantification, each extract was analyzed in triplicate using UPLC. The samples were separated on an Acquity UPLC CSH C18 column, which was 100 mm long and 2.1 mm wide, with 1.7 µm particles (Waters Corporation, Milford, CT, USA), using a 5 µL injection volume. Elution was carried out using a gradient of solvent mixtures A (30% methanol, 50% acetonitrile, 20% water, *v*/*v*) and B (50% methanol, 50% acetonitrile, *v*/*v*) at a flow rate of 0.5 mL/min. An empirically determined elution program was followed. Chromatograms were recorded at specific wavelengths: at 298 nm for γ- and δ-tocopherol, at 293 nm for α-tocopherol homologs, and at 445 nm for carotenoid compounds. Quantification was performed using external calibration curves prepared for each standard compound identified in the extracts. Linear calibration curves (R^2^ > 0.995) were established for all compounds, ensuring accurate quantification. Recovery experiments were conducted by spiking known concentrations of tocopherol and carotenoid standards into blank oil samples. Recovery rates ranged from 92% to 103%, demonstrating negligible matrix effects. The extraction method was validated by analyzing standard reference materials (AOCS Low Erucic Rapeseed Oil, Supelco, Merck KGaA, Darmstadt, Germany). Comparative analyses confirmed the reliability and reproducibility of the procedure, with extraction efficiencies consistently exceeding 95%.

#### 2.2.5. The Radical Scavenging Capacity

The radical scavenging capacity of the lipid samples was evaluated through a DPPH assay on a Genesys 6 spectrophotometer (Thermo Electron Corporation, Waltham, MA, USA), following the methodology described by Tuberoso et al. [20]. To assess the antioxidant potential, a 0.4 mM DPPH solution was prepared by dissolving DPPH (2,2-diphenyl-1-picrylhydrazyl) in ethyl acetate. In an Eppendorf tube, 50 μL of the pumpkin lipid extract was dissolved in 50 μL of ethyl acetate. A 20 μL aliquot of this solution was then transferred to a glass tube containing 3 mL of 0.04 mM DPPH solution in ethyl acetate. The tubes were sealed, mixed, protected from light, and incubated at room temperature for 30 min. Absorbance was measured at 517 nm against ethyl acetate as the blank to quantify the antioxidant activity. A standard curve for Trolox solutions—Trolox being a compound with antioxidant properties used as a reference for the antioxidant potential of the tested samples—was also prepared. The antioxidant potential of the tested samples was expressed as Trolox equivalents per kg of lipid fractions (mM TEAC/kg).

#### 2.2.6. The Oxidative Stability Study

Differential scanning calorimetry (DSC) and the determination of the acidic value (AV) and peroxide value (PV) were utilized to assess the oxidative stability of the lipid samples.

For the DSC method, a Perkin Elmer Pyris 6 calorimeter was used (PerkinElmer, Waltham, MA, USA), based on the methodology outlined by Grajzer et al. [21]. Calibration of the apparatus was performed using indium and zinc. A 3–5 mg lipid sample was placed in an aluminum pan with a layer of ca. 20 mg of aluminum oxide. The pan with the sample was stabilized in the DSC furnace at 30 °C and then exposed to a 50 mL/min oxygen flow. Isothermal tests were conducted at temperatures from 100 to 140 °C, while dynamic analysis ranged from 50 to 350 °C, increasing at 20 °C/min. DSC thermograms depicted downward endothermic peaks, from which kinetic parameters were derived via the Pyris Thermal Analysis integrated software (v10.1).

The determination of the AV and PV was conducted following established protocols as specified by the European Committee for Standardization (CEN) in the years 2008, 2009, and 2010 [22,23]. To measure the AV, 1 g of oil was titrated with potassium hydroxide using phenolphthalein as an indicator to assess the sample’s acidity. The PV was determined by dissolving 0.5 g of oil in a mixture of glacial acetic acid and chloroform, reacting it with potassium iodide, and then titrating with 0.02 mol/L sodium thiosulfate using starch as an indicator.

#### 2.2.7. Statistical Analysis

Statistical analysis of the results was performed using GraphPad Prism 10 (GraphPad Software, San Diego, CA, USA). The obtained results were checked for normal distribution using the Shapiro–Wilk test. Statistical analysis was used based on one-way ANOVA, with Tukey’s post hoc test. The level of significance was set at 5%. The visualization was achieved using the CLD (Compact Letter Display) method, where different letters indicate a significant difference in the comparable means. Principal component analysis (PCA) was performed using the Python 3.12.7 environment. The build-in function from the ‘sklearn’ library was used. The data were normalized by square root transformation and visualized using the ‘matplotlib’ library.

## 3. Results

### 3.1. The Total Fat Content and Fatty Acid Composition

The total fat content in the tested seeds ranged from 51.4% in giant squash, ‘Show Winner’ variety, to 43.11% for the *C. moschata* ‘Butternut Rugosa’ variety (Table 1). The fat derived from the ‘Show Winner’ variety exhibited the most substantial total content of polyunsaturated fatty acids, accounting for 67.5% (Table 1). This elevated level can be attributed primarily to the particularly high proportion of linoleic acid in ‘Show Winner’ fat. Consequently, this extract also manifested an exceptionally high n-6/n-3 fatty acid ratio of 104.9. Notably, an even more pronounced n-6/n-3 ratio was observed in the ‘Moonshine’ fat, reaching 157.5. This heightened ratio can be traced back to the remarkably scant amount of α-linolenic acid identified in the samples.

### 3.2. Phytosterol Composition and Content

In the analyzed pumpkin seeds’ lipid fractions, the overall sterol content was quantified in the range from 469.10 mg/kg in ‘Golden Hubbard’ to 627.26 mg/kg in ‘Jack Sprat’, with sterol contents in extracts from both species represented by a number of varieties falling within a similarly wide span (Table 2). Notably, Δ^7^-avenasterol emerged as the predominant phytosterol, accounting for 29.5% to 42.3% of the total sterol fraction. Furthermore, the lipid extracts exhibited appreciable concentrations of other Δ^7^-sterols, including spinasterol, Δ^7,22,25^-stigmastatrienol, Δ^7,25^-stigmastadienol, and Δ^7^-stigmastenol. Cumulatively, the Δ^7^-sterols represented between 87.8% and 95.4% of the sterol fraction.

### 3.3. Antioxidant Composition and Content

Table 3 summarizes the contents of antioxidants present in the lipid extracts: total and individual tocopherol, carotenoid, and squalene contents.

Significant variations in tocopherol content were noted among the lipid fractions derived from different pumpkin varieties. ‘Moonshine’ seed fat had the lowest tocopherol content at 71.94 mg/kg. In contrast, lipids from three giant pumpkin varieties showed higher tocopherol levels, with ‘Da Marmellata’ fat containing 433.12 mg/kg and ‘Pink Jumbo Banana’ having the highest at 498.37 mg/kg. γ-Tocopherol constituted between 75.2% and 91.7% of the total tocopherols in these pumpkin lipid fractions. Additionally, α- and δ-homologs were detected, varying across the varieties. ‘Show Winner’ fat notably had a high α-tocopherol content of 88.45 mg/kg, while ‘Pink Jumbo Banana’ was rich in δ-tocopherol at 45.01 mg/kg. ‘Moonshine’ seed fat contained a minimal amount of α-tocopherol, at just over 7 mg/kg.

Oil extracted from various pumpkin seed varieties exhibited carotenoid concentrations ranging from 5.68 mg/kg in *C. moschata* to 11.54 mg/kg in *C. maxima* ‘Show Winner’. Xanthophylls such as lutein and zeaxanthin were predominant in most pumpkin seed oils. However, the ‘Golden Hubbard’ lipid fraction was an exception, containing significant amounts of β-cryptoxanthin. The unsaponified fraction of ‘Moonshine’ fat had the highest concentration of lutein 5,6-epoxide among all tested varieties at 1.32 mg/kg; additionally, β-carotene was present at a high amount of 2.96 mg/kg.

The squalene content averaged 943 mg per kg of lipid fraction from the analyzed pumpkin seeds. Notably, the fat derived from *C. pepo* seeds showed the greatest variation in squalene content, with between 635 mg/kg (‘Moonshine’) and 3092 mg/kg (‘Oblonga’). Lipids derived from the ‘Show Winner’ pumpkin variety exhibited the highest squalene concentration of 1928 mg among *C. maxima* representatives. In contrast, the squalene content in the other five lipid fractions from this species varied more modestly: 947 mg/kg for the ‘Pink Jumbo Banana’ variety and 616 mg/kg for ‘Atlantic Giant’, as detailed in Table 3.

### 3.4. Oxidative Stability Studies, Quality Parameters, and Antioxidant Potential

DSC is a widely used technique to investigate oxidation processes by measuring the heat flow associated with phase transitions and chemical reactions as a function of temperature. Parameters such as oxidative induction time (OIT), energy of activation (Ea), and oxidation propagation phase (OPP) are crucial for assessing the oxidative stability and kinetic properties of oils and fats.

Figure 1 and Figure 2 show the OIT and OPP measured at 100 °C, 110 °C, 120 °C, 130 °C, and 140 °C. Among the fats evaluated, the *C. maxima* ‘Pink Jumbo Banana’ lipid fraction demonstrated the highest oxidative stability, as indicated by the most extended oxidation induction times at 100 °C and 110 °C, which were 810.6 and 390.3 min, respectively (Figure 1). Conversely, *C. pepo* ‘Moonshine’ oil exhibited the lowest oxidative stability, with OIT values at these temperatures measuring 407.8 and 178.2 min. Notably, *C. pepo* ‘Moonshine’ fat also had the longest OPP at 100 °C, whereas ‘Pink Jumbo Banana’ fat recorded the shortest. Additionally, the PV of ‘Pink Jumbo Banana’ was significantly the lowest among the studied varieties (Figure 3), distinguishing it as having the most pronounced oxidative stability among the samples tested.

Pumpkin seed lipid fractions exhibited rather low radical scavenging activity, with values ranging from only 0.63 to 1.32 mM TEAC/kg (Figure 3). The lowest activity was observed in the oil from the common pumpkin variety ‘Moonshine’, while the highest was found in the oil from the *C. moschata* variety.

### 3.5. PCA of Lipids Extracted from Various Pumpkin Seed Varieties Based on Nutritional Values

The PCA score plot (Figure 4a) illustrates the separation of various varieties based on their chemical profiles, with Principal Component 1 (PC1) and Principal Component 2 (PC2) together accounting for 81.8% of the total variance. This suggests that the primary chemical differences among these varieties are effectively captured by these two dimensions. Varieties such as ‘Atlantic Giant’ are positioned far to the right, indicating a chemical composition that is markedly distinct from other varieties. *C. pepo* varieties, including ‘Jack Sprat’, ‘Moonshine’, and ‘Mustang’, cluster distinctly on the positive side of PC1, suggesting a unique chemical profile compared to most *C. maxima* and *C. moschata* varieties. These results highlight clear inter-species differences in chemical composition.

The associated loading scatter plot illustrates the contribution of individual compounds to oxidative stability in terms of their positions along PC1 and PC2 (Figure 4b). It seems that components like squalene and total tocopherols, especially γ-tocopherol, are positioned relatively far from the origin and could have a significant influence on oxidative stability. Campesterol and the n-6/n-3 ratio also have a distinct position, indicating a notable contribution to the principal components.

Compounds such as total phytosterols and individual carotenoids (like trans-lutein and all-trans-zeaxanthin) are clustered near the origin in the zoomed-in plot. This close clustering indicates a relatively low influence on oxidative stability or that these compounds have balanced contributions along both components.

## 4. Discussion

There are about 150 cultivars of pumpkins cultivated worldwide, each with varying levels of popularity and unique requirements for climate, soil, and watering regimes. While it is not feasible to study them all, we focused on a selection of diverse varieties representing three species that can be successfully cultivated under the demanding climate of Poland. These cultivars were chosen not only for their ability to produce the highest harvest yields but also for their compatibility with sustainable, green ecological practices. This combination of high-yield potential and environmentally conscious farming makes them particularly suitable for extensive study and application. In this study, we focused on analyzing the oxidative stability of lipid fractions from three pumpkin species, aiming to ascertain which oil components exert, and in what quantitative combinations, the most significant influence on the lipid fraction stability of the studied varieties. Additionally, we sought to identify the most valuable varieties in terms of oxidative stability and nutraceutical properties, providing insights into their suitability for functional and dietary applications.

The studied pumpkin seeds contained an average of 46.4% of fat and were within the upper limits or exceeded the content obtained for pumpkin seeds by other authors [24]. However, the total fat amount obtained in this study was higher than the amount of fat in raw materials such as cotton seeds (20–24%), saffron (30–35%), and soybean (18–22%) and similar to the values obtained for olives and rapeseed (40–48% and 15–50%, respectively) [1]. Due to the high fat content, pumpkin seeds are a food product with a high energy density of approx. 4.5 kcal/g. They can also be an efficient oil raw material. It should be noted, however, that the efficiency of hot *n*-hexane extraction of fat is slightly higher than in the case of extraction by cold pressing, which allows the extraction of 30–40% of fat from pumpkin seeds, according to various data [24]. Pumpkin seeds are rich in unsaturated fatty acids, primarily comprising oleic and linoleic acids. The polyunsaturated fatty acid (PUFA) content in the pumpkin seed fats, represented mainly by linoleic acid, analyzed in our study was considerably elevated compared to pumpkin oils previously studied by other authors. This observation holds true for oils from both *C. maxima* and *C. pepo* species, as well as other species within the *Cucurbita* genus. Those earlier studies reported an average PUFA content of approximately 45% [1,2,4,6,25]. The significant content of linoleic acid (LA) in the tested pumpkin seed oils underscores their value as an outstanding dietary source of this constituent. However, caution is advised in regard to a balanced dietary n-6/n-3 ratio, as it is necessary to also include fats rich in n-3 fatty acids, such as linseed oil and fatty fish from the sea.

Oils recovered from pumpkin seed varieties are abundant in phytosterols, squalene, and tocopherols. The pumpkin seed fats from the 11 varieties, analyzed using gas chromatography coupled with mass spectrometry, contained small amounts of β-sitosterol, a Δ^5^-sterol, which constituted between 2.9% and 9.1% of the sterol fraction. Our findings align with those of other authors who used GC-MS for analysis, reporting similarly low levels of Δ^5^-phytosterols, typically around 5% [26]. However, our findings diverge from some literature data which suggest a more substantial presence of Δ^5^-sterols in pumpkin seed oils [2,27]. These studies indicate that the proportion of Δ^7^-phytosterols does not exceed 76% of the total phytosterol content in industrially produced pumpkin seed oils [2,4]. This may result from the analytical method used, as gas chromatography with HP-1 and HP-5 capillary columns could likely lead to coelution of β-sitosterol and spinasterol [2,28].

The content of squalene in the tested pumpkin seed oils differed significantly from the data in the literature, with some authors indicating very high contents of this component in the pumpkin seed oil of the species studied, even amounting to more than 6.5 g/kg oil [4,29]. According to other authors, the seed oil of *C. pepo* pumpkin varieties contains squalene in the range of 1645–2584 mg/kg, with the highest levels being determined in oils from roasted seeds [30].

The varied tocopherol content in the pumpkin seed fats studied confirms the variability of the tocopherol profile typical for oils from seeds of different *Cucurbita* sp. and varieties. Significant differences in the amounts of these antioxidant compounds have also been noted by other researchers [27,28,31,32]. However, the majority of data suggest a predominant presence of γ-tocopherol in the total tocopherol homologs of pumpkin seed oils [13,27,31,32]. In contrast, Stevenson et al. [6] observed a significant quantitative predominance of the δ form in almost all the *C. maxima* seed oils examined, with these authors generally reporting a much higher total content of these compounds than in the current study. It should be noted that the published data refer to oils extracted using various methods, from either roasted or raw seeds, dehulled or not, which can also affect tocopherol concentrations. For instance, Nakić et al. [32] found different tocopherol contents in oils extracted from intact pumpkin seeds and from those without husks. Roasting is commonly used as a preparatory step in seed pressing [13,33]. According to Vujasinovic et al. [34], the tocopherol content significantly increases with the temperature and duration of this process, suggesting that roasting the pumpkin seeds could further increase the amounts of tocopherols transferred to the extracted oil.

The total carotenoid contents in the pumpkin seed fats analyzed in this study are similar to or lower than the results of carotenoid composition reported by other authors [33]. Parry et al. [12] found a significant presence of zeaxanthin, as well as β-carotene and cryptoxanthin, in the total carotenoids of cold-pressed oil from roasted seeds of the ‘Triple Treat’ cultivar.

Oxidative stability is a crucial quality parameter for edible vegetable oils, as it impacts their applicability in technological processes and determines their shelf life. Various analytical methods are employed to assess the oxidative stability of oils, such as the Rancimat method and active oxygen method (AOM) [24]. In our laboratory practice, we have adopted several methods that allow for the examination of oil oxidation rates. For this work, DSC was utilized due to its rapid response time and efficiency in analyzing reaction kinetics from the onset of oxidation to its termination.

All pumpkin seed oil varieties demonstrated outstanding oxidative stability compared to oils rich in unsaturated fatty acids and limited in ALA, as reported by Tan et al. (2002) [35]. For instance, ‘Show Winner’ (390.3 ± 9.5 min) and ‘Jack Sprat’ (364.28 ± 2.7 min) exhibited a significantly higher OIT at 110 °C than olive oil (169.02 ± 0.82 min), corn oil (166.55 ± 0.18 min), peanut oil (127.21 ± 0.56 min), and canola oil (259.96 ± 0.21 min). Even pumpkin varieties with comparatively lower bioactive contents, such as ‘Moonshine’ (178.2 ± 4.5 min), outperformed traditional unsaturated oils like sunflower oil (131.88 ± 0.76 min) and soybean oil (124.13 ± 1.12 min).

The PCA provides valuable insights into the chemical profiles of the pumpkin varieties’ seeds’ lipid fractions, linking bioactive compound content to oxidative stability and informing their potential applications (Figure 4a,b).

Among the *C. maxima* varieties ‘Pink Jumbo Banana’, ‘Da Marmellata’, and ‘Golden Hubbard’ and the *C. pepo* variety ‘Jack Sprat’ stood out for their distinct chemical profiles, though their roles and applications differ. Additionally, the *C. pepo* variety ‘Moonshine’ has unique characteristics, particularly its remarkably prolonged OPP.

‘Pink Jumbo Banana’ demonstrated superior oxidative stability, evidenced by its high OIT across all tested temperatures. Its PCA position reflects a focused chemical profile, characterized by exceptionally high levels of γ-tocopherol, carotenoids (such as β-carotene and lutein), and squalene, which synergistically enhance antioxidant defense and effectively delay oxidation. The large ratio of γ- and δ-tocopherols to α-tocopherol, amounting to 14.1, previously recognized as a factor positively influencing oil stability, may have further contributed to this effect [36].

‘Pink Jumbo Banana’ demonstrated superior oxidative stability. The expanded PCA provides additional validation, highlighting its higher levels of tocopherols, carotenoids, and squalene as key factors driving its oxidative stability and prolonged OIT. Conversely, ‘Da Marmellata’ clustered slightly higher on the PCA plot, reflecting its broader chemical diversity with a balanced composition of γ-tocopherol, carotenoids, and other bioactives. While its OIT was slightly lower than that of ‘Pink Jumbo Banana’, its diverse antioxidant composition suggests potential for nutraceutical and functional food applications. Notably, the oil from this cultivar exhibited a particularly high ratio of γ- and δ-tocopherols to the α-tocopherol isomer exceeding 20, which could have contributed to its high oxidative stability and may indicate its potential utility as an oxidative stabilizer for oils with a low ratio of these tocopherols [36]. However, its elevated PV indicates greater hydroperoxide formation, highlighting the need for careful formulation and storage to maximize its efficacy.

This study further revealed the synergistic roles of key bioactive compounds in enhancing oxidative stability.

Lipid fractions with an n-6/n-3 ratio between 100 and 170 exhibit the most favorable oxidative stability, exemplified by varieties like ‘Show Winner’, ‘Jack Sprat’, and ‘Golden Hubbard’. These varieties demonstrated a high OIT at 100 °C (OIT100) and a prolonged OPP, whereas ratios above 180, as seen in the ‘Mustang’ variety, did not prove beneficial to the oxidative stability of the oil. These findings align with previous research indicating that higher ratios of n-6 to n-3 fatty acids can increase susceptibility to oxidation [17].

‘Golden Hubbard’ and ‘Jack Sprat’, which clustered closely on the PCA plot, showed remarkably high oxidative stability driven by their unique chemical compositions. Both varieties exhibited a favorable balance of monounsaturated fatty acids (MUFAs), such as oleic acid, which resist oxidation, and polyunsaturated fatty acids (PUFAs), which provide nutritional benefits. Their n-6/n-3 ratios, around 165, further contributed to their stability by reducing their susceptibility to oxidative degradation. Moreover, their high levels of γ-tocopherol (317.91 mg/kg in ‘Golden Hubbard’ and 321.86 mg/kg in ‘Jack Sprat’, Table 3) and squalene (1365.5 mg/kg and 2693.1 mg/kg, respectively) provided robust defense against oxidation.

However, ‘Golden Hubbard’, while not exhibiting the highest γ-tocopherol levels, showed remarkable oxidative stability, largely due to its unique carotenoid profile. The presence of significant amounts of α- and β-cryptoxanthin (1.21 mg/kg and 0.50 mg/kg, respectively) in its lipid fraction suggests a unique mechanism where these xanthophylls complement other antioxidants. Some studies suggest that cryptoxanthin, especially the β form, may stabilize fatty acids against heat degradation [37].

These synergistic effects highlight the potential of the ‘Jack Sprat’ and ‘Golden Hubbard’ varieties for applications requiring high oxidative stability. They are particularly suited for creating shelf-stable functional foods such as nutritional bars or n-3-enriched spreads [38].

The presence of γ-tocopherol shows a clear correlation with an increased OIT at 100 °C, indicating its importance in delaying the onset of oxidation. Varieties such as ‘Show Winner’, ‘Jack Sprat’, and ‘Golden Hubbard’, which are high in γ-tocopherol, demonstrated a strong initial resistance to oxidation. Antioxidant activity operates through the quenching of free radicals, effectively halting the chain reaction that leads to radical accumulation and thereby preventing lipid oxidation. The significance of γ-tocopherol’s role in maintaining oil stability has already been well documented in the literature [8]. However, our data indicate that while tocopherols typically act as antioxidants, high concentrations of total and γ-tocopherols may paradoxically accelerate the oxidative propagation phase in oils rich in LA and OA, indicating potential pro-oxidant activity under specific conditions. This highlights the critical need to balance tocopherol levels relative to the fatty acid composition of oils. This could occur through tocopherols promoting the breakdown of lipid hydroperoxides and fatty acid degradation [32,33].

Interestingly, in soybean oil, with considerable amounts of ALA, γ-tocopherol behaves differently. Studies utilizing advanced techniques like ^1H nuclear magnetic resonance (NMR) demonstrated that increasing the γ-tocopherol concentration delayed the formation of secondary oxidation products, contrary to its pro-oxidant effects shown in our study [39]. This contrast highlights the unique interaction between γ-tocopherol and the ALA-containing fatty acid matrix in soybean oil, where γ-tocopherol exerts stronger antioxidative effects compared to α-tocopherol.

Our findings highlight that γ-tocopherol at a molar ratio of 1:285 to LA provides the most effective stabilization of the lipid fraction [8]. This underscores the importance of maintaining optimal tocopherol levels relative to the fatty acid composition. Furthermore, our data suggest that maintaining a balanced n-6/n-3 ratio of around 100 to 170 is critical for optimizing the stability of pumpkin oils, as excessively high ratios result in diminished benefits. In addition to its primary antioxidant role in stabilizing linoleic acid and delaying the OIT, γ-tocopherol operates synergistically with other bioactive compounds to provide robust oxidative protection, an effect supported by findings from other studies [30].

These insights underscore the dual role of γ-tocopherol as both an antioxidant and, in certain scenarios, a potential pro-oxidant, emphasizing the critical need for balanced bioactive compound concentrations to maximize oil stability. Ratios of tocopherols to linoleic acid below 1:2000 and polyphenols to linoleic acid below 1:3000, as reported in earlier research [8], are insufficient to prevent oxidation in oils dominated by linoleic acid. Our results reinforce the necessity of achieving specific tocopherol concentrations for effective stabilization and long-term oil stability.

Squalene’s position alongside γ-tocopherol in the PCA indicates its synergistic role in enhancing long-term oxidative stability by delaying the propagation phase. This effect was especially pronounced in varieties like ‘Mustang’ (*C. pepo*) and ‘Pink Jumbo Banana’ (*C. maxima*). Squalene’s ability to scavenge oxygen effectively delays oxidation over time [40]. Notably, the stabilizing effect of squalene is most beneficial when combined with other antioxidants, such as γ-tocopherol. ‘Moonshine’ (*C. pepo*) was clustered distinctly on the positive side of PC1, away from most *C. maxima* and *C. moschata* varieties due to its particularly low content of tocopherols and phytosterols among the varieties tested. ‘Moonshine’, while it did not exhibit the highest OIT, stood out for its remarkably prolonged propagation phase. This phase represents the period during which free radicals and hydroperoxides accumulate before secondary products like aldehydes are formed [41]. The deceleration of chain propagation observed in ‘Moonshine’ seed lipids may be attributed to the relatively high concentrations of carotenoids, specifically lutein and zeaxanthin, as well as the limited formation of heat degradation products from tocopherols, which are known to act as triggers for propagation [39].

Although the presented results are intriguing, further studies are necessary to confirm the consistency of the observed relationships using the same cultivars across multiple growing seasons. Additionally, it is essential to conduct investigations based on the framework proposed here, utilizing pumpkin seed oils extracted through more environmentally sustainable and currently promoted methods.

Another important avenue for future research involves evaluating the efficacy of pumpkin seed lipid extracts as additives in blends with fats exhibiting low oxidative stability. Such studies would provide valuable insights into the practical applications of these lipid fractions and their potential to enhance the stability and functionality of nutritionally significant oils.

In reference to the proposed future research directions, it is important to acknowledge several limitations of the present study. These include the relatively small representation of the cultivars analyzed, as well as the evaluation being conducted during a single growing season and at a single location in Poland, which limits the generalizability of the findings. Additionally, while the oil extraction method employed in this study remains widely used and versatile, it is increasingly being replaced by environmentally friendly ‘green’ technologies. Furthermore, although the DSC method used to assess oxidative susceptibility has its advantages, it does not replicate shelf-life conditions and cannot provide real-world data on the durability of oil composition. However, it is important to note that the goal of this study was not to assess the lipid extracts as final market-ready products but rather to explore their potential for developing novel, high-value functional products.

## 5. Conclusions

Our findings demonstrate that a moderate n-6/n-3 ratio, along with balanced levels of γ-tocopherol and squalene, is essential for preserving the oxidative stability of pumpkin seed lipids. Lipid fractions from high-performing Polish *C. maxima* cultivars, such as ‘Show Winner’ and ‘Pink Jumbo Banana’, have emerged as ideal candidates for nutraceutical application due to their elevated levels of tocopherols, carotenoids, and squalene.

Notably, the oxidative stability observed in the pumpkin seed lipids surpassed that of many commonly produced oils, such as sunflower and soybean oils. This stability, driven by unique bioactive profiles, makes pumpkin seed lipids a valuable resource for stabilizing oils with low oxidative resistance and for developing shelf-stable functional foods. Among potential applications is blending with ALA-rich oils such as flaxseed or rose hip seed oils to provide robust antioxidative protection and enhance the oxidative stability.

However, the intervarietal differences are considerable and should be taken into account in designing functional food products. In addition, this diversity allows for harnessing the specific bioactive profiles among the *Cucurbita* species and cultivars to optimize compositions with the desired properties.

Further studies should investigate the consistency of these findings across multiple growing seasons and alternative geographic regions, as well as exploring the use of environmentally sustainable extraction methods to maximize the practical applications of pumpkin seed lipid fractions.

## Figures and Tables

**Figure 1 foods-14-00354-f001:**
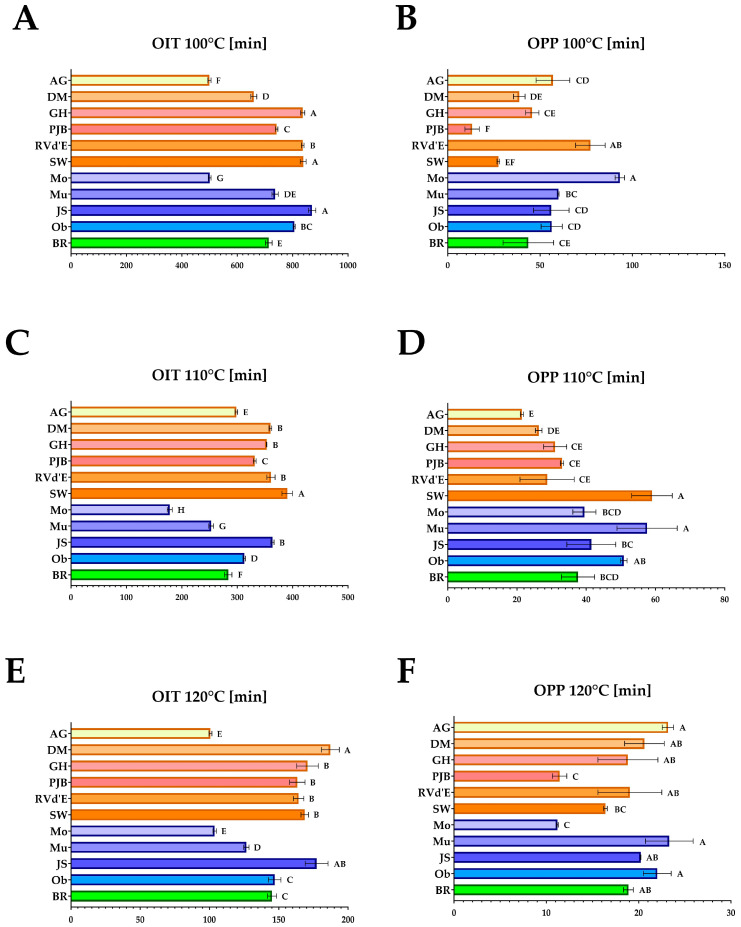
Oxidative stability determined by differential scanning calorimetry: (**A**) Oxidative induction time at constant temperatures of 100 °C [min]; (**B**) Oxidation propagation phase at constant temperature of 100 °C [min]; (**C**) Oxidative induction time at constant temperature of 110 °C [min]; (**D**) Oxidation propagation phase at constant temperature of 110 °C [min] (**E**) Oxidative induction time at constant temperature of 120 °C [min]; (**F**) Oxidation propagation phase at constant temperature of 120 °C [min] ^A,B,C,D,E,F,G,H^—values in the same row that share the same superscript letter are not significantly different; AG—‘Atlantic Giant’; DM—‘Da Marmellata’; GH—‘Golden Hubbard’; PJB—‘Pink Jumbo Banana’; RVd’E—‘Rouge Vif d’Etampes’; SW—‘Show Winner’; Mo—‘Moonshine’; Mu—‘Mustang F1’; JS—‘Jack Sprat F1’; Ob—‘Oblonga’; BR—‘Butternut Rugosa’.

**Figure 2 foods-14-00354-f002:**
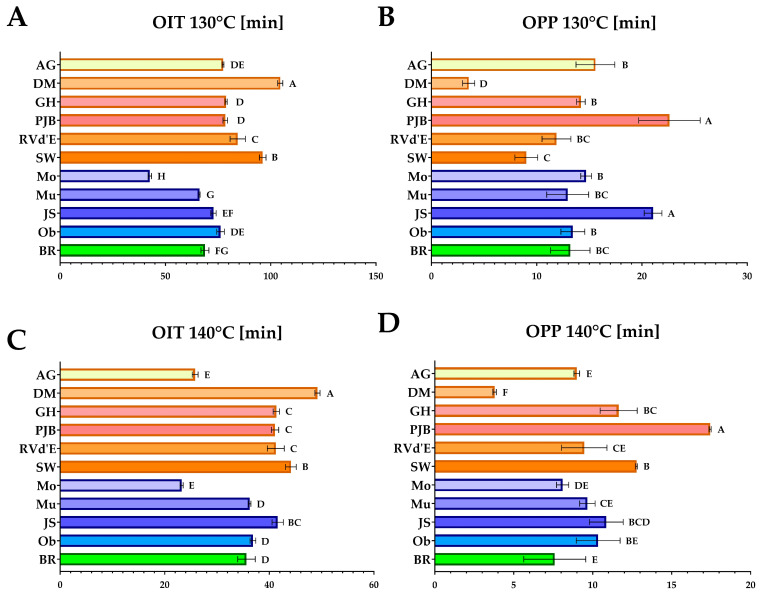
Oxidative stability determined by differential scanning calorimetry; (**A**) Oxidative induction time at constant temperature of 130 °C [min]; (**B**) Oxidation propagation phase at constant temperature of 130 °C [min]; (**C**) Oxidative induction time at constant temperature of 140 °C [min]; (**D**) Oxidation propagation phase at constant temperature of 140 °C [min]; ^A,B,C,D,E,F,G,H^—values in the same row that share the same superscript letter are not significantly different; AG—‘Atlantic Giant’; DM—‘Da Marmellata’; GH—‘Golden Hubbard’; PJB—‘Pink Jumbo Banana’; RVd’E—‘Rouge Vif d’Etampes’; SW—‘Show Winner’; Mo—‘Moonshine’; Mu—‘Mustang F1’; JS—‘Jack Sprat F1’; Ob—‘Oblonga’; BR—‘Butternut Rugosa’.

**Figure 3 foods-14-00354-f003:**
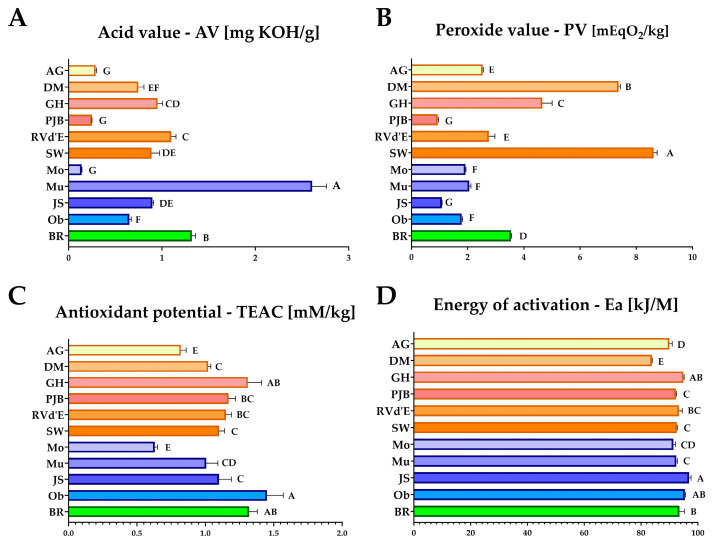
(**A**) Acid value (AV) [mg KOH/g] in lipid fractions from seeds of pumpkin varieties, (**B**) Peroxide value (PV) [mEqO₂/kg] in lipid fractions from seeds of pumpkin varieties, (**C**) Antioxidant potential–DPPH assay (TEAC) [mM/kg] in lipid fractions from seeds of pumpkin varieties, (**D**) Energy of activation (Ea) [kJ/M] in lipid fractions from seeds of pumpkin varieties. ^A,B,C,D,E,F,G^—values in the same row that share the same superscript letter are not significantly different; AG—‘Atlantic Giant’; DM—‘Da Marmellata’; GH—‘Golden Hubbard’; PJB—‘Pink Jumbo Banana’; RVd’E—‘Rouge Vif d’Etampes’; SW—‘Show Winner’; Mo—‘Moonshine’; Mu—‘Mustang F1’; JS—‘Jack Sprat F1’; Ob—‘Oblonga’; BR—‘Butternut Rugosa’.

**Figure 4 foods-14-00354-f004:**
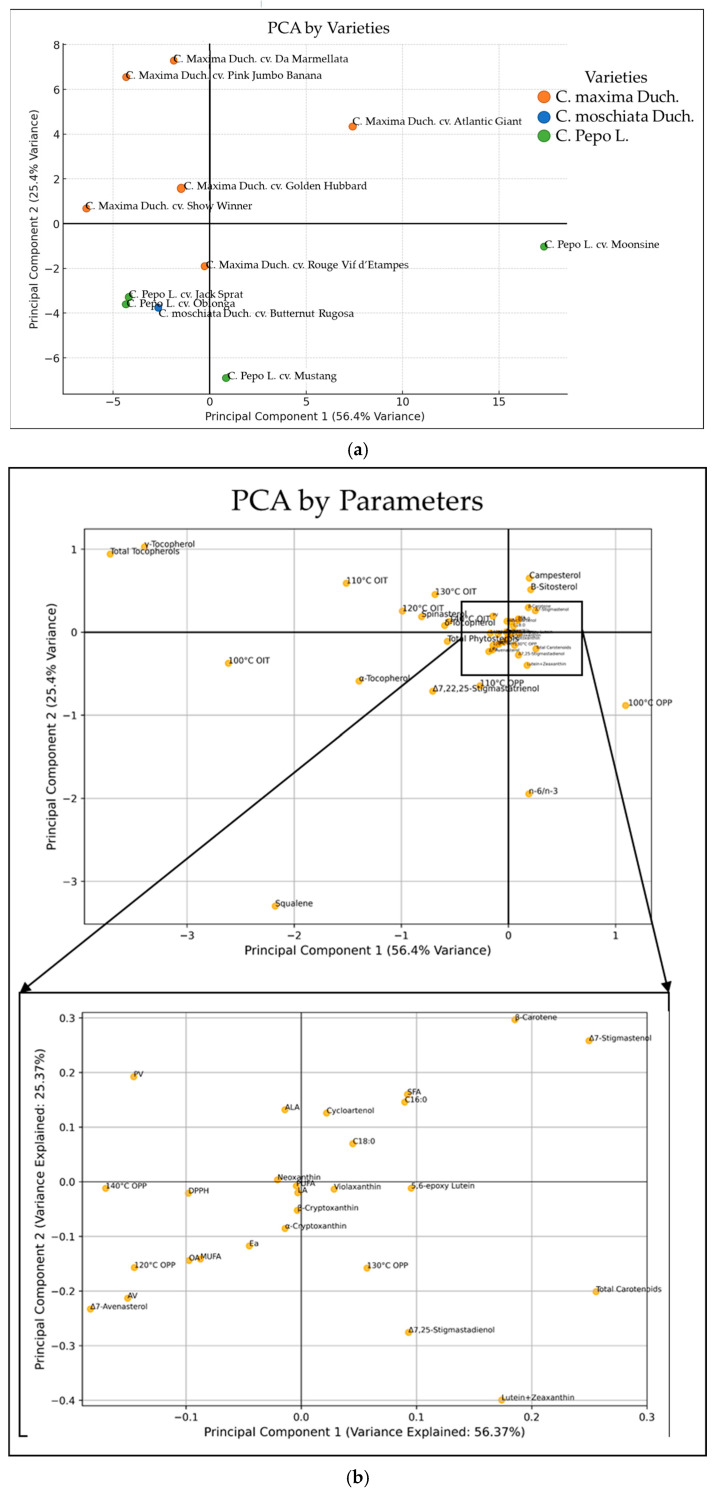
PCA score plot based on the nutritional properties of oils extracted from various pumpkin seed varieties: (**a**) the data displayed segregate the varieties according to their chemical profiles; (**b**) the associated loading scatter plot illustrates the relationships between chemical components and oxidative stability parameters determined in pumpkin seed lipid fractions.

**Table 1 foods-14-00354-t001:** Total lipid content in seeds (%) and fatty acid composition of lipid fractions of eleven pumpkin varieties (%).

Sp.	Variety	Total Lipids [%]	Fatty Acids [%]							
			C16:0	C18:0	C18:1 n-9	C18:2	C18:3 n-3	MUFAs	PUFAs	n-6/n-3
*C. maxima*	AG	48.81 ± 1.50 ^AB^	13.65 ± 0.19 ^C^	5.20 ± 0.05 ^C^	14.65 ± 0.13 ^EF^	65.09 ± 0.34 ^CE^	0.75 ± 0.02 ^B^	14.88 ± 0.19 ^EG^	65.83 ± 0.33 ^BE^	87.13 ± 2.58 ^DE^
	DM	45.95 ± 0.44 ^C^	11.99 ± 0.09 ^EF^	5.64 ± 0.01 ^B^	14.79 ± 0.25 ^E^	65.78 ± 0.32 ^BCD^	0.99 ± 0.05 ^A^	15.14 ± 0.21 ^DE^	66.8 3 ± 0.33 ^ABC^	66.49 ± 3.09 ^E^
	GH	48.13 ± 0.68 ^AB^	14.47 ± 0.05 ^B^	6.27 ± 0.01 ^A^	18.85 ± 0.06 ^B^	59.10 ± 0.01 ^F^	0.36 ± 0.04 ^DE^	19.24 ± 0.07 ^B^	59.46 ± 0.03 ^F^	165.51 ± 20.13 ^BC^
	PJB	46.66 ± 0.58 ^BC^	15.24 ± 0.18 ^A^	5.17 ± 0.11 ^C^	14.00 ± 0.34 ^FG^	64.04 ± 0.58 ^E^	0.79 ± 0.03 ^B^	14.30 ± 0.34 ^G^	64.83 ± 0.60 ^DE^	80.90 ± 4.13 ^DE^
	RVd’E	44.11 ± 2.13 ^C^	11.62 ± 0.03 ^EF^	6.62 ± 0.05 ^A^	14.75 ± 0.07 ^E^	65.94 ± 0.01 ^BC^	0.36 ± 0.02 ^DE^	15.05 ± 0.07 ^EF^	66.32 ± 0.05B ^CD^	183.56 ± 12.65 ^B^
	SW	51.36 ± 2.27 ^A^	12.02 ± 0.42 ^E^	4.28 ± 0.12 ^DE^	15.60 ± 0.32 ^D^	66.82 ± 0.72 ^AB^	0.64 ± 0.03 ^C^	15.77 ± 0.26 ^D^	67.49 ± 0.71 ^AB^	104.94 ± 3.38 ^D^
*C. pepo*	Mo	51.29 ± 1.02 ^A^	14.70 ± 0.17 ^AB^	5.32 ± 0.11 ^BC^	13.89 ± 0.25 ^G^	64.70 ± 0.50 ^CE^	0.41 ± 0.02 ^D^	14.44 ± 0.22 ^FG^	65.16 ± 0.52 ^CE^	157.76 ± 8.24 ^C^
	Mu	44.11 ± 2.12 ^C^	10.20 ± 0.50 ^H^	4.45 ± 0.02 ^DE^	19.26 ± 0.03 ^B^	65.15 ± 0.56 ^BE^	0.31 ± 0.00 ^E^	19.50 ± 0.03 ^B^	65.47 ± 0.57 ^CE^	211.62 ± 2.42 ^A^
	JS	43.13 ± 0.92 ^C^	10.65 ± 0.38 ^GH^	4.53 ± 0.06 ^DE^	19.98 ± 0.30 ^A^	63.72 ± 0.74 ^E^	0.38 ± 0.02 ^D^	20.38 ± 0.30 ^A^	64.1 ± 0.74 ^E^	166.24 ± 8.14 ^BC^
	Ob	43.34 ± 0.67 ^C^	12.80 ± 0.21 ^D^	4.23 ± 0.01 ^E^	17.66 ± 0.31 ^C^	64.18 ± 0.60 ^DE^	0.40 ± 0.00 ^D^	17.87 ± 0.31 ^C^	64.58 ± 0.61 ^DE^	161.54 ± 2.00 ^BC^
*C. moschata*	BR	43.11 ± 0.37 ^C^	11.24 ± 0.40 ^FG^	4.64 ± 0.40 ^D^	14.14 ± 0.24 ^EG^	68.88 ± 0.69 ^A^	0.42 ± 0.01 ^D^	14.38 ± 0.32 ^FG^	68.60 ± 1.18 ^A^	162.24 ± 4.80 ^BC^

^A,B,C,D,E,F,G,H^—values in the same row that share the same superscript letter are not significantly different; MUFAs—monounsaturated fatty acids; PUFAs—polyunsaturated fatty acids; n-6/n-3—ratio of n-6 to n-3 fatty acids; AG—‘Atlantic Giant’; DM—‘Da Marmellata’; GH—‘Golden Hubbard’; PJB—‘Pink Jumbo Banana’; RVd’E—‘Rouge Vif d’Etampes’; SW—‘Show Winner’; Mo—‘Moonshine’; Mu—‘Mustang F1’; JS—‘Jask Sprat F1’; Ob—‘Oblonga’; BR—‘Butternut Rugosa’.

**Table 2 foods-14-00354-t002:** Total phytosterol content and composition [mg/kg].

Sp.	Variety	Phytosterols [mg/kg]								
		Campesterol	Δ^7,22,25^- Stigmastatrienol	β-Sitosterol	Spinasterol	Δ^7,25^-Stigmastadienol	Δ^7^- Stigmastenol	Δ^7^- Avenasterol	24-Methylene-Cycloartenol	Total Sterols
*C. maxima*	AG	10.24 ± 0.45 ^B^	40.63 ± 1.67 ^EF^	37.89 ± 1.42 ^C^	104.83 ± 3.94 ^F^	89.04 ± 3.79 ^A^	43.71 ± 1.10 ^BC^	240.1 ± 9.13 ^B^	2.13 ± 0.01 ^FG^	568.50 ± 18.13 ^B^
	DM	8.38 ± 0.37 ^DE^	34.10 ± 1.50 ^G^	42.41 ± 1.49 ^B^	108.18 ± 3.81 ^EF^	51.89 ± 2.00 ^F^	40.82 ± 1.79 ^C^	185.9 ± 6.61 ^CD^	3.45 ± 0.01 ^AB^	475.13 ± 17.14 ^DE^
	GH	9.87 ± 0.40 ^BC^	36.7 ± 0.50 ^EG^	31.90 ± 1.22 ^D^	139.20 ± 0.17 ^D^	55.75 ± 0.98 ^EF^	30.41 ± 0.02 ^EF^	161.5 ± 0.56 ^F^	3.80 ± 0.03 ^A^	469.10 ± 1.93 ^E^
	PJB	14.46 ± 0.62 ^A^	56.0 ± 2.45 ^D^	51.4 ± 1.03 ^A^	154.32 ± 3.09 ^BC^	74.75 ± 1.40 ^B^	45.72 ± 0.32 ^AB^	167.1 ± 6.52 ^EF^	3.35 ± 0.05 ^B^	567.11 ± 13.17 ^B^
	RVd’E	5.63 ± 0.23 ^F^	41.5 ± 1.84 ^E^	51.71 ± 0.98 ^A^	147.80 ± 1.66 ^BD^	75.88 ± 0.01 ^B^	47.69 ± 0.22 ^A^	172.6 ± 3.37 ^DF^	2.17 ± 0.31 ^EF^	545.00 ± 6.77 ^BC^
	SW	7.62 ± 0.31 ^E^	72.84 ± 1.30 ^B^	18.38 ± 0.28 ^F^	170.52 ± 2.62 ^A^	56.79 ± 0.55 ^EF^	33.05 ± 1.22 ^DE^	265.2 ± 7.32 ^A^	2.82 ± 0.09 ^CD^	627.26 ± 12.38 ^A^
*C. pepo*	Mo	9.00 ± 0.35 ^CD^	34.5 ± 1.68 ^FG^	37.89 ± 0.70 ^C^	99.61 ± 1.84 ^F^	60.99 ± 2.73 ^DE^	43.64 ± 1.83 ^BC^	188.5 ± 9.03 ^CD^	3.13 ± 0.09 ^BC^	477.23 ± 12.73 ^DE^
	Mu	3.03 ± 0.22 ^H^	68.4 ± 3.31 ^BC^	23.4 ± 1.66 ^E^	81.30 ± 0.57 ^G^	91.89 ± 1.25 ^A^	35.81 ± 2.19 ^D^	197.2 ± 3.57 ^C^	2.10 ± 0.10 ^DE^	503.56 ± 11.25 ^DE^
	JS	4.33 ± 0.40 ^G^	71.7 ± 3.88 ^B^	29.30 ± 1.23 ^D^	157.20 ± 4.14 ^B^	79.08 ± 2.65 ^B^	35.42 ± 0.67 ^D^	248.7 ± 5.83 ^AB^	1.78 ± 0.04 ^GH^	627.49 ± 14.47 ^A^
	Ob	2.36 ± 0.06 ^HI^	62.6 ± 0.52 ^C^	39.0 ± 0.44 ^C^	144.60 ± 8.47 ^CD^	64.60 ± 1.92 ^CD^	31.83 ± 0.04 ^E^	233.0 ± 9.10 ^B^	2.53 ± 0.13D ^E^	580.52 ± 13.00 ^B^
*C. moschata*	BR	1.82 ± 0.03 ^I^	91.4 ± 2.56 ^A^	17.90 ± 0.75 ^F^	117.60 ± 2.60 ^E^	68.10 ± 3.38 ^C^	26.89 ± 1.37 ^F^	183.5 ± 3.96 ^CDE^	1.55 ± 0.05 ^H^	508.66 ± 13.05 ^CD^

^A,B,C,D,E,F,G,H,I^—values in the same row that share the same superscript letter are not significantly different; AG—‘Atlantic Giant’; DM—‘Da Marmellata’; GH—‘Golden Hubbard’; PJB—‘Pink Jumbo Banana’; RVd’E—‘Rouge Vif d’Etampes’; SW—‘Show Winner’; Mo—‘Moonshine’; Mu—‘Mustang F1’; JS—‘Jack Sprat F1’; Ob—‘Oblonga’, BR—‘Butternut Rugosa’.

**Table 3 foods-14-00354-t003:** Total tocopherol, carotenoid, and squalene contents [mg/kg] of pumpkin seed fat.

Sp.	Variety	Tocopherols [mg/kg]				Carotenoids [mg/kg]								Squalene [mg/kg]
		α-Tocopherol	γ- Tocopherol	δ-Tocopherol	Total	All-Trans-β-Carotene	All-Trans-Lutein +All-Trans-Zeaxanthin	All-Trans-Neoxanthin	All-Trans-Lutein- 5,6-Epoxid	All-Trans-α-Crypto-Xanthin	All-Trans- β-Crypto- Xanthin	Viola- Xanthin	Total	
*C. maxima*	AG	16.30 ± 2.57 ^D^	217.50 ± 11.55 ^F^	nd	233.84 ± 14.05 ^F^	2.20 ± 0.06 ^B^	6.62 ± 0.18 ^E^	0.25 ± 0.02 ^D^	0.58 ± 0.03 ^C^	0.15 ± 0.00 ^C^	0.19 ± 0.01 ^C^	nd	9.99 ± 0.16 ^F^	616.6 ± 25.1 ^H^
	DM	19.98 ± 1.29 ^D^	397.28 ± 11.36 ^A^	15.87 ± 1.75 ^DE^	433.12 ± 11.59 ^BC^	2.05 ± 0.42 ^B^	3.90 ± 0.25 ^G^	0.48 ± 0.05 ^C^	0.73 ± 0.15 ^BC^	0.08 ± 0.00 ^D^	0.14 ± 0.02 ^D^	nd	7.39 ± 0.05 ^I^	774.4 ± 15.2 ^GH^
	GH	29.74 ± 2.42 ^C^	317.91 ± 14.29 ^CD^	25.90 ± 0.81 ^BC^	373.55 ± 14.05 ^D^	1.29 ± 0.13 ^C^	9.03 ± 0.13 ^C^	0.15 ± 0.04 ^D^	0.31 ± 0.07 ^D^	0.41 ± 0.01 ^B^	1.21 ± 0.00 ^A^	0.50 ± 0.02 ^C^	12.87 ± 0.09 ^C^	1365.5 ± 61.2 ^F^
	PJB	33.17 ± 2.57 ^C^	423.62 ± 18.58 ^A^	45.01 ± 2.37 ^A^	498.37 ± 19.85 ^A^	2.35 ± 0.04 ^B^	4.05 ± 0.04 ^G^	0.75 ± 0.09 ^A^	0.81 ± 0.06 ^B^	0.17 ± 0.00^C^	0.14 ± 0.00 ^D^	nd	8.21 ± 0.05 ^H^	947.3 ± 25.1 ^G^
	RVd’E	20.43 ± 1.44 ^D^	268.49 ± 4.91 ^E^	21.52 ± 0.75 ^CD^	310.44 ± 2.90 ^E^	3.14 ± 0.17 ^A^	6.20 ± 0.48 ^E^	nd	nd	0.84 ± 0.07 ^A^	0.23 ± 0.03 ^BC^	0.48 ± 0.04 ^C^	10.89 ± 0.55 ^E^	2247.9 ± 51.9 ^DE^
	SW	88.45 ± 5.36 ^A^	348.63 ± 16.18 ^BC^	22.39 ± 1.02 ^BC^	466.47 ± 15.05 ^AB^	2.46 ± 0.05 ^B^	7.85 ± 0.03 ^D^	0.59 ± 0.03 ^B^	0.37 ± 0.03 ^D^	0.16 ± 0.02 ^C^	0.11 ± 0.01 ^E^	0.94 ± 0.07 ^B^	11.54 ± 0.08 ^D^	1927.7 ± 33.9 ^E^
*C. pepo*	Mo	7.43 ± 0.35 ^E^	54.08 ± 5.22 ^G^	10.43 ± 0.99 ^E^	71.94 ± 4.94 ^G^	2.96 ± 0.05 ^A^	9.67 ± 0.21 ^B^	0.53 ± 0.02 ^BC^	1.32 ± 0.06 ^A^	0.2 ± 0.00 ^C^	0.25 ± 0.00 ^B^	nd	14.92 ± 0.22 ^B^	634.9 ± 31.0 ^H^
	Mu	41.56 ± 1.84 ^B^	238.97 ± 6.91 ^EF^	nd	280.53 ± 8.53 ^E^	0.41 ± 0.03 ^E^	17.21 ± 0.11 ^A^	nd	nd	nd	nd	1.78 ± 0.09 ^A^	19.40 ± 0.13 ^A^	3175.1 ± 170.1 ^A^
	JS	49.22 ± 4.38 ^B^	321.86 ± 14.47 ^BD^	nd	371.08 ± 18.77 ^D^	0.59 ± 0.03 ^E^	8.03 ± 0.03 ^D^	nd	nd	nd	nd	0.41 ± 0.02 ^C^	9.03 ± 0.06 ^G^	2693.1 ± 28.2 ^C^
	Ob	43.37 ± 0.56 ^B^	357.10 ± 9.09 ^B^	nd	400.48 ± 9.27 ^CD^	0.29 ± 0.04 ^E^	6.49 ± 0.07 ^E^	nd	nd	nd	nd	1.04 ± 0.12 ^B^	7.82 ± 0.13 ^HI^	3092.0 ± 16.8 ^A^
*C. moschata*	BR	45.97 ± 2.54 ^B^	311.04 ± 18.52 ^D^	31.84 ± 3.7 ^B^	388.85 ± 17.87 ^D^	0.36 ± 0.02 ^D^	5.26 ± 0.16 ^F^	nd	nd	nd	nd	0.06 ± 0.00 ^D^	5.68 ± 0.14 ^I^	2889.1 ± 15.2 ^B^

^A,B,C,D,E,F,G,H,I^—values in the same row that share the same superscript letter are not significantly different; AG—‘Atlantic Giant’; DM—‘Da Marmellata’; —‘Golden Hubbard’; PJB—‘Pink Jumbo Banana’; RVd’E—‘Rouge Vif d’Etampes’; SW—‘Show Winner’; Mo—‘Moonshine’; Mu—‘Mustang F1’; JS—‘Jack Sprat F1’; Ob—‘Oblonga’; BR—‘Butternut Rugosa’; nd—not detected.

## Data Availability

The original contributions presented in this study are included in the article. Further inquiries can be directed to the corresponding author.

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
