# Peer review of "Nutraceutical Prospects of Pumpkin Seeds: A Study on the Lipid Fraction Composition and Oxidative Stability Across Eleven Varieties"

_foods, 2025, doi:10.3390/foods14030354_

Round 1

Reviewer 1 Report

Comments and Suggestions for Authors The paper focused on the lipid fraction compositions and oxidative stability of Pumpkin seeds across eleven varieties. The paper is easy to read and meets the scope of the Journal. However, a minor revision of the manuscript is recommended to improve the manuscript. The specific comments are shown below: Line 23, the sentence "The results highlight relations between specific chemical constituents" is invalid. Please rewrite it for clarity.

Lines 154-156, please specify the method of DPPH free radical scavenging capacity.

Lines 330-334, please cite references to support the point.

Line 373, "Differential Scanning Calorimetry (DSC)", it already has an abbreviation in line 236. Please replace it with "DSC".

Author Response

For research article: Nutraceutical Prospects of Pumpkin Seeds: A Study on the Lipid Fraction Composition and Oxidative Stability Across Eleven Varieties

Response to Reviewer #1

We sincerely thank the reviewers for their thorough and constructive evaluation of our manuscript. Your insightful comments and suggestions have provided valuable guidance for refining the clarity, scientific depth, and overall presentation of our work.

We appreciate the time and effort invested in reviewing our work and believe the revisions have resulted in a stronger and more impactful manuscript. Below, we provide a detailed point-by-point response to the comments for your consideration.

Thank you once again for your thoughtful feedback.

Comments and Responses

Comment 1: [Line 23, the sentence "The results highlight relations between specific chemical constituents" is invalid. Please rewrite it for clarity.]
Response 1:
Thank you for pointing out the need for clarity in this sentence. The abstract has been extensively revised, and the sentence in question has been replaced to better reflect the study's findings and scientific intent.

The updated abstract now highlights the intrinsic relationship between the chemical composition of pumpkin seed lipids and their oxidative stability, explicitly focusing on identifying cultivars with significant nutraceutical potential. The revised version emphasizes the evaluation of key chemical components (e.g., fatty acids, sterols, and lipid antioxidants) and their role in enhancing oxidative stability, making the sentence and overall abstract clearer and more aligned with the study's objectives.

We hope this updated version addresses your concerns effectively.

Comment 2: [Lines 154-156, please specify the method of DPPH free radical scavenging capacity.]
Response 2:
Thank you for your valuable comment. The method description has been thoroughly revised and updated as follows:

"The radical scavenging capacity of lipid samples was evaluated through a DPPH assay using a Genesys 6 spectrophotometer (Thermo Electron Corporation, Waltham, Massachusetts, USA), following the methodology described by Tuberoso et al. [18]. To assess antioxidant potential, a 0.4 mM DPPH solution was prepared by dissolving DPPH (2,2-diphenyl-1-picrylhydrazyl) in ethyl acetate. In an Eppendorf tube, 50 μL of the pumpkin lipid extract was dissolved in 50 μL of ethyl acetate. A 20 μL aliquot of this solution was then transferred to a glass tube containing 3 mL of 0.04 mM DPPH solution in ethyl acetate. The tubes were sealed, mixed, protected from light, and incubated at room temperature for 30 minutes. Absorbance was measured at 517 nm against ethyl acetate as the blank to quantify the antioxidant activity. A standard curve for Trolox solutions, a compound with antioxidant properties used as a reference for the antioxidant potential of the tested samples, was also prepared. The antioxidant potential of the tested samples was expressed as Trolox equivalents per kg of lipid fractions (mM TEAC/kg)."

This addition ensures the methodology is described in sufficient detail for reproducibility.

Comment 3: [Lines 330-334, please cite references to support the point.]
Response 3:
Thank you for your suggestion. The missing citations have been incorporated to substantiate the discussed point, as follows:

  • Rezig, L.; Chouaibi, M.; Msaada, K.; Hamdi, S. Chemical composition and profile characterization of pumpkin (Cucurbita maxima) seed oil. Industrial Crops and Products, 2012, 37, 82–87. DOI: https://doi.org/10.1016/j.indcrop.2011.12.004
  • Vorobyova, O.; Bolshakova, A.; Pegova, R.; Kol’chik, O.; Klabukova, I.; Krasilnikova, E.; Melnikova, N. Analysis of the components of pumpkin seed oil in suppositories and the possibility of its use in pharmaceuticals. Journal of Chemical and Pharmaceutical Research, 2014, 6, 1106–1116.

These references provide adequate support for the statement and enhance the academic rigor of the content.

Comment 4: [Line 373, "Differential Scanning Calorimetry (DSC)", it already has an abbreviation in line 236. Please replace it with "DSC".]
Response 4:
Thank you for your observation. The term "Differential Scanning Calorimetry (DSC)" has been replaced with "DSC" as per your suggestion to maintain consistency, as the abbreviation was already introduced in line 236.

Reviewer 2 Report

Comments and Suggestions for Authors

Present work aimed to characterize the lipid profile of 11 samples of three species of pumpkin. Different etraction methods and GC, GC-MS and HPLC were used for lipid identification, DSC, DPPH were used to measured the antioxidant activity of samples.

Materials.

Lines 70-79, this paragraog is not clear: Eleven pumpkin cultivars of 3 species... The seedlings were obtained from certified seeds and were representing from the following varieties.

here.. authors get the seeds from botanical garden or seedlings?, the authors let grow seedlings until pumpkin was obtained, then seeds were collected?

please clarify which are the varieties corresponding to the three species

How many replicates were used?

Line 73.. verities... should be varieties?

Line 151-152... the content was determined using the external calibration curve corresponding to each identified standard... Not clear this part.. it was used a calibration with with identified standars?

Minor comments:

the use of oC.. sometimes uses underlined, min or minutes

et al.  or et al.;  or sometimes as; Line 156 Tuberoso et al (18) or sometimes as line 162,, Grahzer et al. (2023)(19)

Line. 169-171, please explain briefy the methods

Table 1 should be as soon it was mentioned in the text

Line 241-247.. where I can see these info?, there is a figure or table?

Table is very big,

the last line TAEC (which it was not mentioned in materiales and methods).. should be in another table or figure,

the same in tablle 2. the values AV and PV, could be another figure or table

letters in Figure 1 are not visible to read.

Line 342, calls to Table 12

Lines 353 and 407, the style of reference is not according to the journal

Tables are very big

which includes different info; lipids and antioxidant,

as recommended, maybe antioxidant values should be presented in a graphic? or different table,

Comments on the Quality of English Language

some engllish corrections is needed

Author Response

For research article: Nutraceutical Prospects of Pumpkin Seeds: A Study on the Lipid Fraction Composition and Oxidative Stability Across Eleven Varieties

Response to Reviewer #2

We sincerely thank the reviewers for their valuable feedback and constructive comments on our manuscript. Your insights have greatly helped us improve the clarity, depth, and overall quality of our work. Below, we provide a detailed response to each comment, highlighting the revisions made to address your suggestions. Thank you for your time and effort in reviewing our submission.

Comment 1: [Lines 70-79, this paragraph is not clear. Clarify whether the authors used seeds from a botanical garden or seedlings. Specify varieties for each species and describe replication methods.]

Response 1:
Thank you for your comment. We have clarified this section to enhance its readability and ensure transparency in the experimental setup:
Eleven pumpkin cultivars representing three species (Cucurbita maxima Duchesne, C. moschata Duchesne, and C. pepo L.) were studied. Seeds were obtained from certified sources and corresponded to the following varieties:

  • Cucurbita maxima: ‘Atlantic Giant,’ ‘Da Marmellata,’ ‘Golden Hubbard,’ ‘Pink Jumbo Banana,’ ‘Rouge Vif d’Etampes,’ ‘Show Winner.’
  • Cucurbita moschata: ‘Butternut Rugosa.’
  • Cucurbita pepo: ‘Moonshine,’ ‘Mustang F1,’ ‘Jack Sprat F1,’ ‘Oblonga.’

The seeds were planted in outdoor plots at the Botanical Garden of Adam Mickiewicz University in Poznań, Poland, under common garden conditions. The plants were cultivated to maturity, and the ripe fruits were manually harvested. Seeds were extracted, cleaned of pulp, and dried in a glasshouse without heating, maintaining a maximum temperature of 24°C.

The study utilized three biological replicates for each pumpkin variety to ensure the reliability of results. These details have been updated in the manuscript to provide a clear description of the experimental procedures.[line 93-106, Material section, p.2]

Comment 2: [Line 73: "verities" should be corrected to "varieties."]

Response 2:
Thank you for catching this error. The term "verities" has been corrected to "varieties" to ensure accuracy.[line 95,page 3]

Comment 3: [Line 151-152: Clarify whether external calibration curves were used with identified standards.]

Response 3:
Thank you for your comment. The text has been revised for clarity and now states:
" Quantification was performed using external calibration curves prepared for each standard compound identified in the extracts."[193-195]

This ensures that the method used for quantification is clearly explained.

Comment 4: [Minor formatting issues: "°C," "min," and "minutes."]

Response 4:
Thank you for pointing out these inconsistencies. All instances of "°C," "min," and "minutes" have been reviewed and standardized throughout the manuscript for consistency.

Comment 5: [Formatting inconsistencies in "et al." references.]

Response 5:
We appreciate your observation. All citations have been reviewed and updated to ensure they follow the correct journal format. For example, "Tuberoso et al (18)" and "Grahzer et al. (2023)(19)" have been corrected for consistency.

Comment 6: [Lines 169-171: Provide a brief explanation of the methods for AV and PV determination.]

Response 6:
Thank you for your feedback. The description of the methods for Acidic Value (AV) and Peroxide Value (PV) determination has been expanded for clarity:

  • AV was determined by titrating 1 g of oil with potassium hydroxide using phenolphthalein as an indicator to assess acidity.
  • PV was measured by reacting 0.5 g of oil with potassium iodide in a mixture of glacial acetic acid and chloroform, followed by titration with 0.02 mol/dm³ sodium thiosulfate using starch as an indicator.

These changes ensure that the methodology is adequately detailed. [Line 232-236]

Comment 7: [Table 1 should be placed immediately after it is mentioned in the text.]

Response 7:
We have adjusted the manuscript to ensure that Table 1 is placed immediately after its first mention, improving the flow and alignment of the manuscript.

Comment 8: [Lines 241-247: Clarify where the referenced information is presented.]

Response 8:
The referenced information is now presented in Figure 1 and 2. This ensures the data is visually accessible and linked clearly to the text. We added following phrase to point out where the referenced information is presented. ‘Figures 1 and 2 show the OIT and OPP measured at 100°C, 110°C, 120°C, 130°C and 140°C. ‘[line 301-302]

Comment 9: [Table 1 is too large.]

Response 9:
To improve readability, we have divided Table 1 into smaller, focused tables. Additionally, some results from Table 2 have been transferred to a chart for better clarity and data interpretation.

Comment 10: [TAEC is not mentioned in the methods and should be presented separately.]

Response 10:
We have now described the method for assessing Total Antioxidant Equivalent Capacity (TAEC) in the Materials and Methods section. Additionally, TAEC results are presented in a newly created Figure 3 for better context and clarity.

Comment 11: [AV and PV values should be in a separate figure or table.]

Response 11:
Thank you for your suggestion. In response, we have created Figure 4 to present AV and PV values separately. This change ensures better organization and clarity in presenting the data.

Comment 12: [The text in Figure 1 is not visible.]

Response 12:
Figure 1 has been updated with improved resolution and enhanced text readability to meet journal standards.

Comment 13: [Line 342, calls to Table 12.]

Response 13:
Thank you for spotting this error. Upon review, we have reorganized Table 1 and Table 2, ensuring they are properly aligned with their corresponding mentions in the text. Additionally, some data from these tables have been transferred into figures to improve readability and visual representation.

This reorganization enhances the clarity and structure of the manuscript, making it easier for readers to follow the results.

Comment 14: [References in Lines 353 and 407 do not follow the correct format.]

Response 14:
Thank you for pointing this out. All references have been reviewed and updated to adhere to the journal’s citation style.

We trust that these revisions address all your comments comprehensively. Thank you again for your valuable feedback.

Reviewer 3 Report

Comments and Suggestions for Authors

The data of the paper is not enough to support the views of the paper, the logic is poor, and the innovation is insufficient. It is not recommended for publication in food, and there are many issues that need to be resolved in the manuscript.

1、Line 77-78, “medium hight temperatures”, please indicate the specific temperature. Line 79, “freeze-dried”, why not choose the same drying method for the samples? Dehulled or not?

2、Line 83, “freeze-dried seeds (50 g)”, whole seeds or crushed seeds?

3、Please, check the format of tables according to journal requirements, such as Table 2.

4、Figure1 should be given as clear picture.

5、Line 342, Table 12?

6、The interaction of chemical components in different varieties of pumpkin seeds and their effects on oxidative stability and nutritional quality have not been fully discussed.

7、Revise carefully reference style.

Author Response

For research article: Nutraceutical Prospects of Pumpkin Seeds: A Study on the Lipid Fraction Composition and Oxidative Stability Across Eleven Varieties

Response to Reviewer#3

We sincerely thank the reviewer for their constructive feedback on our manuscript. Your comments provided valuable insights and helped us address important areas of improvement, including data presentation, logical flow, and scientific innovation. Below, we provide a detailed response to your concerns, along with the specific revisions made to strengthen the manuscript.

Comment 1: The data of the paper is not enough to support the views of the paper, the logic is poor, and the innovation is insufficient. It is not recommended for publication in Food, and there are many issues that need to be resolved in the manuscript.

Response 1:
We appreciate your feedback and have made significant revisions to the manuscript to address these concerns:

  1. Improved Data Presentation:
    • We reorganized and divided larger tables into smaller, more focused tables to improve readability and data interpretation.
    • Additionally, some data have been converted into figures and charts for clearer visual representation and to facilitate better comprehension.
  2. Enhanced Logical Flow:
    • The discussion section has been expanded to provide a more in-depth analysis of the results, incorporating comparisons with existing studies and emphasizing the unique findings of the study.
    • Critical evaluations of the chemical interactions within pumpkin seed oils and their impact on oxidative stability have been thoroughly discussed.
  3. Increased Innovation:
    • We have integrated new comparative data highlighting the oxidative stability of pumpkin seed oils in relation to other oils rich in unsaturated fatty acids (e.g., sunflower, sesame, olive, soybean, and flaxseed oils). This comparison demonstrates the superior oxidative properties of pumpkin seed oils and their potential industrial applications.
    • We also incorporated dietary implications of the n-6/n-3 fatty acid ratio in pumpkin seed oils, providing practical insights into their use as a complement to other dietary fats.

These revisions significantly enhance the scientific depth, innovation, and logical structure of the manuscript.

Comment 2: [Line 77-78: “medium high temperatures”, please indicate the specific temperature.]

Response 2:
Thank you for your comment. The text has been revised to specify the exact temperature:
"The seeds were manually extracted from ripe pumpkin fruits, manually cleaned of pumpkin pulp, and dried under a roof in a glasshouse without heating, at ambient temperatures not exceeding 24°C."[Line 105-106, p.3]

This revision ensures clarity and provides specific drying conditions.

Comment 3: [Line 79: “freeze-dried”, why not choose the same drying method for the samples?]

Response 3:
The freeze-drying method was selected to ensure the complete removal of residual water and preserve thermolabile bioactive compounds (e.g., tocopherols, carotenoids). This method minimizes degradation and maintains the lipid fractions' integrity, which is crucial for accurate analysis.

This explanation has been added to the manuscript to clarify our rationale.[Line115-117,p/3]

Comment 4: [Dehulled or not?]

Response 4:
The seeds were dehulled before lipid extraction. This information has been added to the Methods section (2.2.1):
"Seeds were dehulled, and lipids were then extracted with n-hexane..." [Line 110,p.3]

Comment 5: [Line 83: “freeze-dried seeds (50 g)”, whole seeds or crushed seeds?]

Response 5:
The seeds used in the study were crushed seeds that had been freeze-dried. This clarification has been explicitly stated in the revised manuscript to avoid ambiguity.[Line 111-112, p.3]

Comment 6: [Please check the format of tables according to journal requirements, such as Table 2.]

Response 6:
The formatting of Table 2 has been updated to comply with the journal's requirements. Additionally, some data previously presented in tables have been converted into figures for better clarity and improved presentation.

Comment 7: [Figure 1 should be a clear picture.]

Response 7:
Figure 1 has been updated to improve resolution and readability, meeting the journal’s standards.

Comment 8: [Line 342: Table 12?]

Response 8:
Thank you for spotting this error. Upon review, we have reorganized Table 1 and Table 2, aligning them properly with their mentions in the text. Some data from these tables have also been converted into figures to enhance readability and structure.

Comment 9: [The interaction of chemical components in different varieties of pumpkin seeds and their effects on oxidative stability and nutritional quality have not been fully discussed.]

Response 9:
We have expanded the discussion to address this comment comprehensively:

  1. PCA Analysis:
    We included a detailed description of the PCA analysis, highlighting how bioactive compounds (e.g., tocopherols, carotenoids, and squalene) contribute to the oxidative stability of different pumpkin seed varieties. For example, we discussed how ‘Pink Jumbo Banana’ and ‘Show Winner’ demonstrated exceptional stability due to their high tocopherol and carotenoid content, while ‘Moonshine’ exhibited prolonged propagation phases due to high carotenoid content compensating for lower tocopherol levels.
  2. Comparative Oxidative Stability:
    We compared the oxidative induction times (OIT) of pumpkin seed oils with those of other oils rich in unsaturated fatty acids (e.g., olive, sunflower, sesame, and soybean oils) based on literature (e.g., Tan et al., 2002). Pumpkin seed oils consistently outperformed these oils, showcasing their exceptional stability.
  3. Dietary Implications of the n-6/n-3 Ratio:
    • We evaluated the dietary implications of the high n-6/n-3 ratio in pumpkin seed oils, recommending their use as a complement to n-3-rich sources (e.g., flaxseed oil) to achieve a balanced fatty acid intake.
    • Additionally, we identified the optimal n-6/n-3 ratio range (100–170) for oxidative stability in pumpkin seed oils.

These points have been added to the revised manuscript, supported by six new citations to enhance the depth and scientific rigor of the discussion.

Comment 10: [Revise reference style carefully.]

Response 10:
All references have been reviewed and updated to adhere to the journal's formatting requirements.

We trust that these revisions adequately address the reviewer’s comments. The manuscript has been significantly improved in terms of data presentation, logical flow, and scientific innovation. We appreciate your time and effort in reviewing our work and look forward to your feedback.

Round 2

Reviewer 2 Report

Comments and Suggestions for Authors

Authors have improved the manuscript, no further comments

Author Response

Comment 1 [Authors have improved the manuscript, no further comments]

Response 1 [Thank you for taking the time to review our revised manuscript and for your positive feedback. We are pleased to hear that the improvements made in response to the first round of comments have addressed your concerns, and we appreciate your acknowledgment that no further comments or revisions are required.

Your valuable input has greatly contributed to enhancing the quality and clarity of our work, and we are grateful for your efforts throughout the review process.]

Reviewer 3 Report

Comments and Suggestions for Authors

1、Line 195, R2.

2、Please, check the format of tables (Table 12 and 3) according to journal requirements.

3、Figure 2, the horizontal coordinate is missing a title and the unit format is wrong.

4、Figure 4(b) should be given as clear picture.

Author Response

Response to Reviewer Comments (Round 2)

Dear Reviewer,

Thank you for your continued evaluation of our manuscript and for providing additional constructive feedback. We have carefully addressed all comments and made the necessary revisions to improve the quality and clarity of the manuscript. Below are our responses to each comment:

Comment 1:[Line 195, R2.]

Response 1:[Thank you for your observation. "R2" has now been formatted with the proper superscript notation as "R²" to reflect the correct scientific representation. We appreciate your remark, which has helped improve the clarity and precision of the manuscript.

If further adjustments are needed, we are happy to address them promptly.

Comment 2:[Please, check the format of tables (Table 1, 2, and 3) according to journal requirements.]

Response:
All tables have been carefully reviewed and reformatted to meet the journal's requirements. Specifically:

  • Table 1: Adjusted to ensure proper alignment, consistent font size, and inclusion of footnotes where applicable.
  • Table 2: Reformatted to ensure the data is clearly presented with uniform column widths and appropriate captions.
  • Table 3: Updated to align with the journal's style guide, including modifications to spacing, headers, and explanatory notes for clarity.
    These changes improve the visual consistency and readability of the tables.

Comment 3:

[Figure 2, the horizontal coordinate is missing a title and the unit format is wrong.]

Response:
The horizontal axis title for Figure 2 has been added to clearly specify the variable represented. Additionally, the unit format has been corrected to align with scientific conventions and the journal's requirements. The updated figure ensures accurate representation and clarity for readers.

Comment 4:

[Figure 4(b) should be given as a clear picture.]

Response:
Figure 4(b) has been updated with a high-resolution image to enhance its clarity and readability. The new figure meets the journal's requirements for resolution and format, ensuring that the data is visually accessible and easy to interpret.

Additional Revision:
In response to feedback, we have also significantly improved the Conclusion section of the manuscript. The revised conclusion provides a clearer and more concise summary of the study’s findings and their implications, better aligning with the overall scope and contribution of the research.

We appreciate the thorough review and constructive feedback, which have significantly enhanced the quality of our manuscript. The revised manuscript, updated figures, and tables have been uploaded for your review. If further clarification or adjustments are needed, we are happy to address them promptly.

Thank you again for your time and effort.